# Changes in Tobacco Use Patterns during COVID-19 and Their Correlates among Older Adults in Bangladesh

**DOI:** 10.3390/ijerph18041779

**Published:** 2021-02-12

**Authors:** Sabuj Kanti Mistry, Armm Mehrab Ali, Md. Ashfikur Rahman, Uday Narayan Yadav, Bhawna Gupta, Muhammad Aziz Rahman, Rumana Huque

**Affiliations:** 1Department of Health Research, ARCED Foundation, 13/1, Pallabi, Mirpur-12, Dhaka 1216, Bangladesh; mehrabbabu@gmail.com; 2Centre for Primary Health Care and Equity, University of New South Wales, Sydney, NSW 2052, Australia; unyadav1@gmail.com; 3BRAC James P Grant School of Public Health, BRAC University, 68 Shahid Tajuddin Ahmed Sharani, Mohakhali, Dhaka 1212, Bangladesh; 4Global Research and Data Support, Innovations for Poverty Action, 101 Whitney Avenue, New Haven, CT 06510, USA; 5Development Studies Discipline, Khulna University, Khulna 9208, Bangladesh; ashfikur@ku.ac.bd; 6Center for Research, Policy and Implementation, Biratnagar 56613, Nepal; 7Department of Public Health, Torrens University, Melbourne, VIC 3000, Australia; bhawna.gupta@laureate.edu.au; 8School of Health, Federation University Australia, Berwick, VIC 3806, Australia; ma.rahman@federation.edu.au; 9Australian Institute of Primary Care and Ageing, La Trobe University, Melbourne, VIC 3086, Australia; 10Department of Economics, University of Dhaka, Dhaka 1000, Bangladesh; rumanah14@yahoo.com; 11Research and Development, ARK Foundation, Gulshan, Dhaka 1212, Bangladesh

**Keywords:** tobacco use, smoking, smokeless tobacco, COVID-19, Bangladesh

## Abstract

The present study explored the changes in tobacco use patterns during the COVID-19 pandemic and their correlates among older adults in Bangladesh. This cross-sectional study was conducted among 1032 older adults aged ≥60 years in Bangladesh through telephone interviews in October 2020. Participants’ characteristics and COVID-19-related information were gathered using a pretested semi-structured questionnaire. Participants were asked if they noted any change in their tobacco use patterns (smoking or smokeless tobacco) during the COVID-19 pandemic compared to pre-pandemic (6 months prior to the survey). Nearly half of the participants (45.6%) were current tobacco users, of whom 15.9% reported increased tobacco use during the COVID-19 pandemic and all others had no change in their tobacco use patterns. Tobacco use was significantly increased among the participants from rural areas, who had reduced communications during COVID-19 compared to pre-pandemic (OR = 2.76, 95%CI:1.51–5.03). Participants who were aged ≥70 years (OR = 0.33, 95% CI: 0.14–0.77), widowed (OR = 0.36, 95% CI: 0.13–1.00), had pre-existing, non-communicable, and/or chronic conditions (OR = 0.44, 95% CI: 0.25–0.78), and felt themselves at the highest risk of COVID-19 (OR = 0.31, 95% CI: 0.15–0.62), had significantly lower odds of increased tobacco use. Policy makers and practitioners need to focus on strengthening awareness and raising initiatives to avoid tobacco use during such a crisis period.

## 1. Introduction

Tobacco use is one of the leading causes of premature death and morbidity [1]. The World Health Organization (WHO) reported that around 1 billion people are currently smokers, of whom 80% are living in low-middle income countries (LMICs). It has been estimated that tobacco causes about 8.8% of total annual global deaths (4.9 million) and 4.1% of disability-adjusted life years (DALYs) (59.1 million) [2]. 

All countries globally have been affected by tobacco, but now this epidemic is shifting from high-income countries to LMICs including Bangladesh, a South Asian country, because of the international expansion of major tobacco companies [3]. Smoking tobacco and consuming smokeless tobacco (SLT) products are significant risk factors of non-communicable diseases (NCDs) [4]. A total of 126,000 deaths were caused by tobacco-related diseases in 2018 in Bangladesh [5]. According to the most recent Global Adult Tobacco Survey, 35.3% of adults were consuming tobacco products in Bangladesh in 2017, where the prevalence of smoking was 18.0%. Notably, the smoking prevalence was considerably higher among males [6]. While people of all ages are at risk of tobacco use, the prevalence of tobacco use was the highest among older adults aged 55 years and above (64.2%; smoking: 26.9%; SLT: 47.1%) in Bangladesh [7], making them at risk of tobacco-related morbidities and death.

The COVID-19 pandemic is the most remarkable public health calamity of this century. The disease continues to invade the entire world; as of 28 January 2021, there were more than 2 million deaths with more than 100 million confirmed cases worldwide. More than 533 thousand confirmed cases with 8072 deaths were also reported by the same date in Bangladesh [8]. Although everybody is at risk of COVID-19, certain groups of the population are more vulnerable than others to its deadliest effects. One such group is older adults who are particularly vulnerable to morbidity and mortality from COVID-19, as they often suffer from different non-communicable and/or chronic conditions such as diabetics, hypertension, and obesity [9,10,11]. Evidence suggests that hospitalisation rates are higher among older adults aged 65 years or over, and deaths are 23 times more likely compared to those aged below 65 years [12]. 

At the same time, the growing pandemic is augmenting existing mental health problems by exacerbating the extent of loneliness, anxiety, panic, depression, and long-term psychosocial impacts among older adults [13]. Prolonged lockdown, social distancing, and self-isolation are increasing stress and fear, with the potential upsurge of tobacco use [14]. In addition, tobacco use may increase due to its use as a coping strategy for the increasing levels of anxiety and stress exacerbated by the isolation and quarantine imposed during COVID-19 [11,15]. The diverse nexus of intertwined biological and socio-ecological factors amid this pandemic alongside the lack of access to health care services may augment the density of depression, suicide, domestic violence, and psychiatric illnesses [10], resulting in a higher use of tobacco products.

While no clear relationship has been established so far between smoking and COVID-19, evidence suggests that tobacco use increases the severity of the disease among COVID-19 patients [14,16,17]. During the previous Middle East respiratory syndrome coronavirus (MERS-CoV) outbreak as well, smokers were twice as likely to be affected by and incur deaths from the influenza than non-smokers [18,19]. Therefore, the present study was carried out in Bangladesh to know about the changes in tobacco use patterns during COVID-19 among older adults. The study also explored the factors associated with changes in tobacco use among older adults in Bangladesh.

## 2. Materials and Methods

### 2.1. Study Design and Participants

This cross-sectional study was conducted remotely through telephone interviews in October 2020. We utilised a pre-established registry, developed through merging the contact information of households from ten different community-based studies accomplished by Aureolin Research, Consultancy, and Expertise Development (ARCED) Foundation during 2016–2020 as sampling frame. The registry comprised almost 9000 households for which verified contact information was available. It included households from mixed demographic groups, including urban and rural areas, different income groups, and households from all eight administrative divisions of Bangladesh.

The sample size of 1096 was calculated considering a 50% prevalence with a 5% margin of error, at the 95% level of confidence, 90% power of the test, and 95% response rate. However, only 1032 participants participated in the final study, with an overall response rate of approximately 94%. To ensure representativeness from all divisions, we adopted a probability proportionate to the number of older adults in each division [20].

A stratified random sampling technique was followed to select the targeted households from each administrative division. One eligible respondent was interviewed from each of the sampled households. In cases where there was more than one eligible participant in a selected household, the oldest one was interviewed. Daily attempts to reach the sampled households by phone were made for up to three days. It required an average of 1.30 (SD ±0.57) phone calls for each participant to successfully survey the 1032 participants. In cases where any household from a certain administrative division was not possible to reach, did not fit the inclusion criteria, or refused to participate, a replacement sample was taken randomly from that specific administrative division. The inclusion criterion was being aged 60 years and above, and the exclusion criteria included severe mental health problems (clinically proved schizophrenia, bipolar mood disorder, dementia/cognitive impairment), a hearing disability, extensive memory problems resulting in the unreliability of answers, or inability to communicate.

### 2.2. Measures

#### 2.2.1. Outcome Measures

The outcome variable of this study considered whether the current tobacco users (smoking any tobacco products or using any SLTs within 30 days of the survey [21]) perceived any change in their tobacco use patterns during COVID-19 compared to before the pandemic (6 months prior to the survey). We asked the participants if they have noted any change in their tobacco use patterns during COVID-19. We categorised and coded the outcome variables as changes in tobacco use patterns during COVID-19 (0 = no change, 1 = increased). We did not find anyone who reported to decrease tobacco use during the COVID-19 pandemic; thus, this change was excluded from the coding.

#### 2.2.2. Explanatory Variables

The explanatory variables considered in this study were as follows: age (categorised as 60–69, 70–79, and ≥80); sex (male/female); marital status (married/widowed); literacy (illiterate/literate); family size (≤4 or >4); family income in Bangladeshi Taka (BDT) (<5000, 5000–10,000, >10,000); residence (urban/rural); occupation (currently employed/unemployed or retired); living arrangements (living alone or with family); memory or concentration problems (no problem/low memory or concentration); presence of pre-existing non-communicable chronic conditions (yes/no); concerned about COVID-19 (hardly, sometimes/often); difficulty in getting food, medicine, and routine medical care during COVID-19 (yes/no); frequency of communication with friends and family during COVID-19 (less than previous/same as previous); and source of COVID-19-related information (TV/radio, health workers, and friends/family/neighbours).

Self-reported information on pre-existing medical conditions, such as arthritis, hypertension, heart diseases, stroke, hypercholesterolemia, diabetes, chronic respiratory diseases, chronic kidney disease, and cancer were also collected.

#### 2.2.3. Data Collection Tools and Techniques

A validated semi-structured questionnaire in the Bengali language, in which most of the questions were adopted from previous studies, was used to collect the information [22,23,24]. The Bengali version of the tool was piloted among a small sample (n = 10) of older adults to refine the language in the final version. The participants approved the tool designed by research team without any corrections or suggestions.

Data collection was conducted through telephone interviews by 10 data collectors and the responses were noted in the SurveyCTO mobile app (Dobility, Inc., Washington, DC, USA) (https://www.surveycto.com/). The data collectors were trained extensively for four days by Sabuj Kanti Mistry, ARM Mehrab Ali, and Uday Narayan Yadav remotely using the Zoom platform (Zoom Video Communications, Inc., San Jose, CA, USA) before the data collection.

#### 2.2.4. Statistical Analysis

The distribution of the variables was assessed through descriptive analysis. Given our variables’ categorical nature, Chi-square tests were performed to compare differences in the changes in tobacco use by explanatory variables, with a 5% level of significance. We used binary logistic regression models to explore the factors associated with dichotomised changes in tobacco use. The initial model was run with all potential covariates, then using the backward elimination criteria with the Akaike information criterion (AIC), a final model was selected. The adjusted odds ratio (aOR) and associated 95% confidence interval (95% CI) were reported. All analyses were performed using the statistical software package Stata (Version 14.0, StataCorp LLC, Texas, TX, USA).

#### 2.2.5. Ethical Approval

The study protocol was approved by the institutional review board of Institute of Health Economics, University of Dhaka, Bangladesh (Ref: IHE/2020/1037). Verbal informed consent was sought from the participants before administering the survey. Participation was voluntary, and participants did not receive any compensation.

## 3. Results

A total of 1032 older adults participated in this study, of whom 471 participants (45.6%) were using tobacco, i.e., smokers (20.2%), SLT users (33.1%), and dual users (7.7%). Of the 471 current tobacco users, 21.0% were from the Dhaka division, 77.1% aged 60–69 years, 72.0% male, 72.0% from the rural areas, 57.5% illiterate, 82.8% were married, and 93.4% were residing with their family members. Over half of them (61.6%) had a family income of >10,000 Bangladeshi Taka (1 Bangladeshi Taka ≈ 0.012 USD) and resided at more than 30 min walking distance from the nearest health centre (53.1%). Details are shown in Table 1.

### 3.1. Changes in Tobacco Use during COVID-19

The study revealed that none of the participants started tobacco use during this pandemic, and no current smokers reported a decrease in their tobacco use. However, 15.9% of the participants reported that their tobacco use had increased during COVID-19. Tobacco smoking increased among 13.5% of the participants who were current smokers, while the use of SLT products increased among 15.2% of the current SLT users. No significant difference was observed between male and female participants in terms of increased tobacco use during COVID-19. A significantly higher percentage of participants (*p* < 0.001) from rural areas reported that they had increased frequency of tobacco use during the COVID-19 pandemic compared to those from urban areas (19.8% vs. 6.1%). Similarly, the percentage with increased tobacco use was significantly higher among the participants who had less communication with others during this pandemic than previously (25.4% vs. 10.4%). The frequency of increased tobacco use was also significantly higher (*p* < 0.05) among participants aged 60–69 years, married, participants from poor households, those from rural areas, participants with pre-existing non-communicable chronic conditions, those who were hardly overwhelmed by COVID-19, and those who did not receive any financial support during COVID-19. See Table 1 for details.

### 3.2. Factors Associated with Changed Tobacco Use during COVID-19

In the adjusted model, age, marital status, residence, pre-existing non-communicable chronic conditions, perceived risk of COVID-19, and frequency of communication during COVID-19 as source of COVID-19-related information were all significantly associated with increased frequency of tobacco use among the participants (Table 2).

We found that the participants who were aged 70 years or more had 67% lower odds of increased tobacco use than those aged 60–69 years (aOR = 0.33, 95% CI: 0.14–0.77), and married participants had 64% lower odds of increased tobacco use than those who were widowed (aOR = 0.36, 95% CI: 0.13–1.00). Participants residing in rural areas had nearly three times higher odds of increased tobacco use than those residing in urban areas (aOR = 2.97, 95% CI: 1.27–6.94), and participants who had less frequency of communication with others during the pandemic had nearly three times higher odds of increased tobacco use frequency (aOR = 2.76, 95% CI: 1.51–5.03). On the other hand, participants who had one or more pre-existing non-communicable chronic conditions had 56% lower odds (aOR = 0.44, 95% CI: 0.25–0.78), and those who felt themselves at the highest risk of COVID-19 had around 70% lower odds (aOR = 0.31, 95% CI: 0.15–0.62) of increased tobacco use.

## 4. Discussion

To the best of our knowledge, this is the first study carried out among older adults in Bangladesh to explore the change in tobacco use patterns amid the COVID-19 pandemic. The present study noted an overall 15.9% rise of tobacco use in older Bangladeshi adults amid this COVID-19 pandemic. The frequency of smoking increased among 13.5% of the participants who were currently smokers, while the frequency of using smokeless tobacco products increased among 15.2% of the participants. This percentage is relatively higher compared to a recent study carried out among the adult population in Belgium, which reported that 7.5% of the participants had an increased frequency of smoking during COVID-19 [25]. Another study also explored the tobacco use patterns in five countries (Italy, India, South Africa, the United Kingdom, and the United States) during COVID-19 and noted an increased frequency of tobacco use during COVID-19 [14].

We found that a significantly higher percentage of participants from rural areas reported that they had increased tobacco use during this pandemic compared to those from urban areas. Several studies demonstrated that tobacco use is significantly higher among rural inhabitants than those of urban areas [26,27]. However, the reasons behind the increased frequency of tobacco use among rural participants compared to those in urban areas during this pandemic remain unclear. One probable reason could be that rural inhabitants were more prone to be distressed and anxious during the COVID-19 pandemic because of inadequate health care facilities, limited capabilities for public protection, and the unavailability of appropriate information [28,29], which might have led to increased tobacco use among the participants. Moreover, rural respondents are more dependent on other income-earning members who mainly work in the informal sector [30]; thus, they might have greater anxiety about future job losses of family members. This combined with the low literacy rate among older adults [31] could also lead to increased tobacco use.

Our study also demonstrated that the participants who had less frequency of communication with others during COVID-19 had the highest odds of increased tobacco use. As the present study was carried out during the peak of COVID-19 in Bangladesh, older adults having less communication with their friends and family members might have felt a loneliness that contributed to an increase in risky behaviour, such as more frequent tobacco use than that of normal times, which was also reported in several other studies [32,33,34]. Evidence suggests that emotions of sadness and loneliness increase the likelihood of depressive symptoms and can increase the consumption of tobacco products in the process [18]. Several other studies also documented that smoking increases with increased mental illness [35,36]. This is also an important finding, not only from the tobacco control policies viewpoint, but also from the health promotion perspective, as the health awareness campaigns for the current pandemic, or for any health-related issue, may not reach this group due to a lack of communication. Hence, alternate platforms such as community meetings could be a potential alternative to improve awareness in that vulnerable cohort.

Our study revealed that increase in tobacco use was significantly lower among participants aged 70 years or older compared to their younger counterparts. We also found that participants who had any pre-existing non-communicable and/or chronic conditions had a lower chance of increased tobacco use during the pandemic. This is possibly because relatively older participants and those who were suffering from any non-communicable and/or chronic conditions were more concerned with the deadliest effects of COVID-19 and thus had lower tobacco use. Similarly, our results also suggest that the participants who perceived that they were at the highest risk of being infected with COVID-19 had significantly lower odds of increased frequency of tobacco use. As argued previously, these participants probably became more aware of the fact that COVID-19 seriously damages the lungs with different respiratory illnesses, and tobacco use can aggravate the situation to a great extent [10,11,14,19]. Indeed, several studies demonstrated that tobacco use can lead to more adverse COVID-19-related outcomes than those of non-smokers [37,38], which could provide a greater motivation to decrease tobacco use. Therefore, giving up tobacco use would potentially aid efforts to tackle not only smoking-related chronic diseases, but also the new infectious-disease threat of COVID-19 [17,25]. Although a recent study argued that risky behaviour increases with increased stress related to the perceived vulnerabilities of COVID-19 [15], this was not the case in our study.

While LMICs are lagging behind in promoting tobacco control compared to developed countries, Bangladesh is one of the countries among the LMICs that has implemented tobacco control policies since 1990 [26]. This has been achieved with different measures such as banning smoking in public places and transports and banning the advertising and promotion of tobacco and tobacco product vending machines, in addition to requiring graphic warning labels on all tobacco products. The warnings provide general statements about the health consequences of tobacco use, stating that smoking causes death, lung cancer, stroke, heart disease, respiratory problems, or other problems. The text warnings cover up at least 50% of the front and back of cigarette and bidi packs [27]. In addition, the Bangladesh government initiated a policy to increase the price and taxes for all kinds of cigarettes and tobacco products, including upward adjustment tax rates [26,27]. Nevertheless, the present study revealed that the frequency of consumption of tobacco products increased significantly during the COVID-19 pandemic. Therefore, policymakers and public health professionals need to consider delivering information about risky behaviours, such as tobacco use, while disseminating the health promotion messages in Bangladesh during this crisis of COVID-19. The involvement of health workers can augment the process in this regard [39]. This study’s findings and policy implications are also very relevant for other countries with similar socio-economic backgrounds.

The limitations of the study include selection bias, as the sampling frame was based on the available household-level information in our registry. Second, there are potential possibilities of recall bias, as the study participants were aged 60 years and above. Third, due to the cross-sectional design, causality cannot be established. Furthermore, a strong limitation of our study is that all the information was self-reported and was based on subjective perceptions. On the other hand, the target sample size was achieved within the period of the pandemic, which indicated a strength of this study, as we had significant power to test our hypotheses.

## 5. Conclusions

We found that tobacco use increased as a coping strategy to manage stress and anxiety among older people of ≥60 years in Bangladesh amid the COVID-19 pandemic. Residing in rural areas and having less communication with others were both significant predictors for increased tobacco use in this study. Policy makers and public health practitioners need to consider strengthening awareness and raising initiatives utilising accessible channels to reach this vulnerable cohort of the population. These initiatives are needed not only to control tobacco use, but also to disseminate health awareness messages including the current regular communications about COVID-19, specifically for those living in rural areas, to help them to quit tobacco during such a pandemic. Research is also required to test the effectiveness and cost-effectiveness of such interventions. 

## Figures and Tables

**Table 1 ijerph-18-01779-t001:** Participants’ characteristics and bivariate analysis of changes in tobacco use during COVID-19.

Characteristics	Overall (n = 1032)	Tobacco User (n = 471)	Change in Tobacco Use		
			No Change	Increased	
	n (%)	n (%)	n (%)	n (%)	*p*
Overall			396(84.1)	75(15.9)	
Division					
Barishal	149(14.4)	62(13.2)	49(79.0)	13(21.0)	0.556
Cottogram	137(13.3)	62(13.2)	54(87.1)	8(12.9)	
Dhaka	210(20.4)	99(21.0)	79(79.8)	20(20.2)	
Mymensingh	63(6.1)	25(5.3)	20(80.0)	5(20.0)	
Khulna	158(15.3)	72(15.3)	64(88.9)	8(11.1)	
Rajshahi	103(10.0)	44(9.3)	36(81.8)	8(18.2)
Rangpur	144(14.0)	69(14.7)	61(88.4)	8(11.6)	
Sylhet	68(6.6)	38(8.1)	33(86.8)	5(13.2)	
Age (year, %)					
60–69	803(77.8)	363(77.1)	297(81.8)	66(18.2)	0.014
≥70	229(22.2)	108(22.9)	99(91.7)	9(8.3)	
Sex					
Male	676(65.5)	339(72.0)	286(84.4)	53(15.6)	0.783
Female	356(34.5)	132(28.0)	110(83.3)	22(16.7)	
Marital status					
Married	840(81.4)	390(82.8)	320(82.1)	70(18.0)	0.008
Widow/Widower	192(18.6)	81(17.2)	76(93.8)	5(6.2)	
Literacy					
Illiterate	602(58.3)	271(57.5)	231(85.2)	40(14.8)	0.422
literate	430(41.7)	200(42.5)	165(82.5)	35(17.5)	
Family size					
≤4	318(30.8)	134(28.5)	115(85.8)	19(14.2)	0.514
>4	714(69.2)	337(71.6)	281(83.4)	56(16.6)	
Family income (BDT)					
<5000	145(14.1)	60(12.7)	43(71.7)	17(28.3)	0.001
5000–10,000	331(32.1)	121(25.7)	96(79.3)	25(20.7)	
>10,000	556(53.9)	290(61.6)	257(88.6)	33(11.4)	
Residence					
Urban	269(26.1)	132(28.0)	124(93.9)	8(6.1)	0.000
Rural	763(73.9)	339(72.0)	272(80.2)	67(19.8)	
Current occupation					
Employed	419(40.6)	236(50.1)	201(85.2)	35(14.8)	0.516
Unemployed	613(59.4)	235(49.9)	195(83.0)	40(17.0)	
Living arrangement					
Living with other family members	953(92.3)	440(93.4)	372(84.6)	68(15.5)	0.295
Living alone	79(7.7)	31(6.6)	24(77.4)	7(22.6)	
Dependent on family for living					
No	329(31.9)	146(31.0)	119(81.5)	27(18.5)	0.307
Yes	703(68.1)	325(69.0)	277(85.2)	48(14.8)	
Walking distance to the nearest health centre					
<30 min	508(49.2)	221(46.9)	183(82.8)	38(17.2)	0.478
≥30 min	524(50.8)	250(53.1)	213(85.2)	37(14.8)	
Problem in memory or concentration					
No problem	782(75.8)	360(76.4)	305(84.7)	55(15.3)	0.490
Low memory or concentration	250(24.2)	111(23.6)	91(82.0)	20(18.0)	
Pre-existing chronic conditions					
No	424(41.1)	162(34.4)	122(75.3)	40(24.7)	0.000
Yes	608(58.9)	309(65.6)	274(88.7)	35(11.3)	
Concerned about COVID-19					
Hardly	299(29.0)	118(25.1)	99(83.9)	19(16.1)	0.951
Sometimes/often	733(71.0)	353(74.9)	297(84.1)	56(15.9)	
Overwhelmed by COVID-19					
Hardly	370(36.4)	142(30.7)	117(78.0)	33(22.0)	0.014
Sometimes/often	647(63.6)	321(79.3)	279(86.9)	42(13.1)	
Feeling himself at highest risk of COVID-19					
No	603(58.4)	292(62.0)	234(80.1)	58(19.9)	0.003
Yes	429(41.6)	179(38.0)	162(90.5)	17(9.5)	
Difficulty in getting food during COVID-19					
No	553(55.3)	221(48.0)	186(84.2)	35(15.8)	0.629
Yes	447(44.7)	239(52.0)	205(85.8)	34(14.2)	
Difficulty in getting medicine during COVID-19					
No	733(75.3)	322(71.6)	275(85.4)	47(14.6)	0.782
Yes	240(24.7)	128(28.4)	108(84.4)	20(15.6)	
Difficulty receiving routine medical care during COVID-19					
No	751(72.8)	311(66.0)	254(81.7)	57(18.3)	0.047
Yes	281(27.2)	160(34.0)	142(88.8)	18(11.2)	
Feeling of loneliness					
Hardly	822(79.7)	366(77.7)	301(82.2)	65(17.8)	0.042
Sometimes/often	210(20.3)	105(22.3)	95(90.5)	10(9.5)	
Frequency of communication during COVID-19					
Same as previous	598(58.0)	298(63.3)	267(89.6)	31(10.4)	0.000
Less than previous	434(42.0)	173(36.7)	129(74.6)	44(25.4)	
Received financial support during COVID-19					
No	764(74.0)	323(68.6)	264(81.7)	59(18.3)	0.040
Yes	268(26.0)	148(31.4)	132(89.2)	16(10.8)	

**Table 2 ijerph-18-01779-t002:** Factors associated with increased tobacco use during COVID-19.

Characteristics	AOR	95% CI
Age (year, %)		
60–69	Ref	
≥70	0.33	0.14–0.77
Marital status		
Married	Ref	
Widow/Widower	0.36	0.13–1.00
Family size		
≤4	Ref	
>4	1.67	0.87–3.18
Family monthly income (BDT)		
<5000	Ref	
5000–10,000	0.89	0.37–2.15
>10,000	0.69	0.29–1.66
Residence		
Urban	Ref	
Rural	2.97	1.27–6.94
Current occupation		
Employed	Ref	
Unemployed or retired	1.36	0.76–2.46
Pre-existing conditions		
No	Ref	
Yes	0.44	0.25–0.78
Feeling themselves at the highest risk of COVID-19		
No	Ref	
Yes	0.31	0.15–0.62
Frequency of communication during the pandemic		
Same as previous	Ref	
Less than previous	2.76	1.51–5.03
Receiving COVID-19-related information from friends/family/neighbours		
No	Ref	
Yes	0.31	0.16–0.61

AOR = Adjusted odds ratio; Note: the model was adjusted for all the variables in the table.

## Data Availability

The data are available upon reasonable request from the corresponding author.

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
