# Peer review of "Changes in Tobacco Use Patterns during COVID-19 and Their Correlates among Older Adults in Bangladesh"

_ijerph, 2021, doi:10.3390/ijerph18041779_

Round 1
Reviewer 1 Report
In this study, the authors used data collected from about 1000 elderlies in Bangladesh through telephone interviews to investigate changes in smoking patterns during COVID-19 and associated factors. The authors found high prevalence of tobacco use (45.6%), 15.9% of which increased tobacco use amid COVID-19. The authors identified several factors associated with the increase in tobacco use: no pre-existing conditions, low perceived risk of COVID-19, less frequent communications with friends and family, etc.
Overall, this is a worthwhile topic to investigate. It is policy-revenant and this paper provides additional information on tobacco use during the COVID-19 pandemic. I have a few comments as follows:
- The authors sampled individuals/households from a pre-established registry of ARCED Foundation. I’d like to see more information on the registry and more discussion the representativeness of the sampling frame and on how it will bias the findings.
- Results in Table 2 suggested that respondents who were overwhelmed by COVID-19 were less likely to increase tobacco use. I find this to be inconsistent with “Frequency of increased tobacco use was also significantly higher (P<0.05) among participants … who were overwhelmed by COVID-19 (line 223-227).
- Also, line 233-238: “In the adjusted model, marital status, residence, overwhelmed by COVID-19, … and friends/family/neighbours as source of COVID-19 related information were significantly associated with increased frequency of tobacco consumption among the participants (Table 3).” Results on “overwhelmed by COVID-19” and “friends/family/neighbours as source of COVID-19 related information”, however, were not reported in Table 3.
- The manuscript would benefit from a more in-depth discussion about the significance of the findings, in particular the contribution to the literature on tobacco use and COVID-19 pandemic.
Author Response
|
Comment/Suggestion |
Authors’ Responses |
|
In this study, the authors used data collected from about 1000 elderlies in Bangladesh through telephone interviews to investigate changes in smoking patterns during COVID-19 and associated factors. The authors found high prevalence of tobacco use (45.6%), 15.9% of which increased tobacco use amid COVID-19. The authors identified several factors associated with the increase in tobacco use: no pre-existing conditions, low perceived risk of COVID-19, less frequent communications with friends and family, etc. I have a few comments as follows: |
Thank you for your comment. |
|
1.The authors sampled individuals/households from a pre-established registry of ARCED Foundation. I’d like to see more information on the registry and more discussion the representativeness of the sampling frame and on how it will bias the findings. |
Thank you for the comment. More details on the sampling frame and sampling strategy are added in the Methods section. Please see page 3 line 137-143 and page 4 line 152-166 |
|
2.Results in Table 2 suggested that respondents who were overwhelmed by COVID-19 were less likely to increase tobacco use. I find this to be inconsistent with “Frequency of increased tobacco use was also significantly higher (P<0.05) among participants … who were overwhelmed by COVID-19 (line 223-227). |
Thank you for pointing out the typo. It was supposed to be “…who were hardly overwhelmed by COVID-19”. We have made the change in the manuscript (page 9 line 271) |
|
3.Also, line 233-238: “In the adjusted model, marital status, residence, overwhelmed by COVID-19, … and friends/family/neighbours as source of COVID-19 related information were significantly associated with increased frequency of tobacco consumption among the participants (Table 3).” Results on “overwhelmed by COVID-19” and “friends/family/neighbours as source of COVID-19 related information”, however, were not reported in Table 3. |
These were included by mistake. Thanks for pointing them out. We have edited this in the revised manuscript. Please see page 10 line 278-281 |
|
4.The manuscript would benefit from a more in-depth discussion about the significance of the findings, in particular the contribution to the literature on tobacco use and COVID-19 pandemic. |
Thanks for your suggestion. We have revised the Discussion and Conclusion section based on your suggestion. |
Reviewer 2 Report
This manuscript presents the findings of a survey in Bangladesh concerning changes in cigarette consumption during the COVID-19 pandemic. I have the following major suggestions:
- The introduction is far too long and should be focused on the question addressed.
- Are there any data on sales of cigarettes or of smokeless tobacco products that could be assessed for trends?
- The paper's analysis is unnecessarily lengthy. The key finding is the percentage increasing tobacco use and also that no one decreased tobacco use. The extensive listing of correlates is not helpful, particularly given data limitations and sample size. A straightforward analysis of factors associated with the change in smoking on an a priori basis might be provided.
- What is the impact of the finding for tobacco control and messaging?
- Are the findings relevant elsewhere?
Author Response
|
Comment/Suggestion |
Authors’ Responses |
|
This manuscript presents the findings of a survey in Bangladesh concerning changes in cigarette consumption during the COVID-19 pandemic. I have the following major suggestions: |
|
|
1.The introduction is far too long and should be focused on the question addressed. |
Thanks for your comments. We have streamlined the introduction based on the research question in the revised manuscript. |
|
2.Are there any data on sales of cigarettes or of smokeless tobacco products that could be assessed for trends? |
Thanks for your comments. Unfortunately, we don’t have this information. |
|
3. The paper's analysis is unnecessarily lengthy. The key finding is the percentage increasing tobacco use and also that no one decreased tobacco use. The extensive listing of correlates is not helpful, particularly given data limitations and sample size. A straightforward analysis of factors associated with the change in smoking on an a priori basis might be provided. |
|
|
4. What is the impact of the finding for tobacco control and messaging? |
We have added the following lines: “Policy makers and public health practitioners need to consider strengthening awareness raising initiatives utilizing accessible channels to reach this vulnerable cohort of population, not only to control tobacco use but also disseminating health awareness messages including the current regular communication of COVID-19, specifically for those living in the rural areas, to help them quit tobacco during such pandemic.” Please see page 14 line 437-444 |
|
5. Are the findings relevant elsewhere? |
Thanks for noting this. We have added the following line: “The study findings and policy implications are also very much relevant for other countries with similar socio-economic background.” Please see page 13 line 415-417 |
Reviewer 3 Report
line 51: "disability-adjusted life yearS"
line 60-61: smoking tobacco and consuming smokeless tobacco products are risk factors (not cause of death)
line 70-71: all ages are at risk (not only the highest for prevalence of tobacco use)
line 98: I suggest to move the locationing of Bangladesh ("a South-Asian country") in the first time you mention it in the manuscript (line 59)
line 130: you should better clarify how do you sample household (simple or stratified sampling - for which variable...). Did you interview all the older adults of the household (or only one)?
line 133: did you apply over-sampling to reach the estimated number (1096) or did you replace participants in case of refuse?
line 135: how many call (mean) did you need for participants?
line 147: it is very surprising that none of 1032 participants reported decreasing in tobacco use...neither positive to Covid-19? Did you ask if they were postive to Covid-19 before the interview?
line 151: was "unmarried" possible?
line 155: you should explain the reason for considering "memory or concentration problems"; did you exclude participants that have memory problems for unreliability of answers?
line 191: is the "istitutional review board" expert of ethical aspects? Is it an Ethics Committee?
line 217 and elsewhere: I suggest to use "tobacco use" instead of "tobacco consumption"
line 228: Delete "Please"
Table 1 and 2: you may put together results of table 1 and 2 obtaining one table with 5 columns so we can have in one sight descriptive and bivariate analysis of all variables. Did you explore correlations between variables?
Table 3: 95%CI are enough, you can omit p value
line 241: delete widowed data that were reported below in line 243
line 273: participants who had "SOME" pre-existing
line 275-278: this is in contrast with the next phrase
line 278-281: this is in contrast with the next phrase
line 291-292: it could be explained by different income and level of literacy in rural and urban areas
line 296-298: why did you provide this information here?
line 344: another strong limitation is that all information are self-reported so are based on subjective perceptions
line 346-347: this conclusion is not well supported by results (you can only describe association not causality), I sugget to modify
line 369: add that the consent was verbal
Author Response
|
Comment/Suggestion |
Authors’ Responses |
|
line 51: "disability-adjusted life yearS" |
Thank you. Fixed in the manuscript. Please see page 2 line 51. |
|
line 60-61: smoking tobacco and consuming smokeless tobacco products are risk factors (not cause of death) |
Thanks for noting this. We have corrected this in the revised manuscript. Please see page 2 line 62-63. |
|
line 70-71: all ages are at risk (not only the highest for prevalence of tobacco use) |
Thanks for noting this. We have included this in the revised manuscript. Please see page 2 line 70-71. |
|
line 98: I suggest to move the locationing of Bangladesh ("a South-Asian country") in the first time you mention it in the manuscript (line 59) |
Thank you. Added in the manuscript. Please see page 2 line 60. |
|
line 130: you should better clarify how do you sample household (simple or stratified sampling - for which variable...). Did you interview all the older adults of the household (or only one)? |
Thanks for your comment. More details on the sampling frame and sampling strategy are added in Methods section. Please see page 3 line 137-143 and page 4 line 152-166 |
|
line 133: did you apply over-sampling to reach the estimated number (1096) or did you replace participants in case of refuse? |
In case any household from a certain administrative division was not possible to reach, did not fit the inclusion criteria, or refused to participate, a replacement sample was taken randomly from that specific administrative division. We have added this in the manuscript. Please see page 4 line 159-162 |
|
line 135: how many call (mean) did you need for participants? |
Following lines are added: “The sampled households were attempted up to three different days to reach over the phone. It required on an average of 1.30 (SD ±0.57) number of phone calls for each participant to successfully survey 1032 participants.” Please see page 4 line 156-159 |
|
line 147: it is very surprising that none of 1032 participants reported decreasing in tobacco use...neither positive to Covid-19? Did you ask if they were postive to Covid-19 before the interview? |
Thanks for noting this. We asked the participants if they were COVID-19 positive before the interview, but less than 1% responded that they had COVID-19 before the interview. This can be attributed to low number testing, not willingness to test and so forth. |
|
line 151: was "unmarried" possible? |
There was no case as unmarried. |
|
line 155: you should explain the reason for considering "memory or concentration problems"; did you exclude participants that have memory problems for unreliability of answers? |
Thanks very much for your information. The relationship between tobacco use pattern and COVID-19 can be linked with how the information related to COVID-19 is being processed by the participants. For example, if someone listens to the importance of reducing tobacco use during COVID-19 and forgets after sometimes due to memory or concentration problem might not consider that advice of changing tobacco use. We have included this in the manuscript. Please see page 4-5, line 200-206
We have excluded the participants who were suffering from extensive memory problems that results in unreliability of answers. Please see page 4 line 162-166 |
|
line 191: is the "istitutional review board" expert of ethical aspects? Is it an Ethics Committee? |
Yes, it is an ethics committee. They have experts in their team on ethical aspects and meet on a regular basis to assess the research proposals and relevant tools on ethical grounds. You can find more details on the following website:
https://ihe.ac.bd/irb/
|
|
line 217 and elsewhere: I suggest to use "tobacco use" instead of "tobacco consumption" |
Thanks for noting this. We have edited the manuscript as you have suggested. |
|
line 228: Delete "Please" |
Deleted, thank you. |
|
Table 1 and 2: you may put together results of table 1 and 2 obtaining one table with 5 columns so we can have in one sight descriptive and bivariate analysis of all variables. |
Thanks for your suggestion. We have merged table 1 and table 2 into table 1. |
|
Did you explore correlations between variables? |
|
|
Table 3: 95%CI are enough, you can omit p value |
Thanks for your suggestion. P values are now omitted from table 3 which is now table 2 in the revised manuscript. |
|
line 241: delete widowed data that were reported below in line 243 |
Thanks for noting this. We have removed this. Please see page 10 line 286-287 |
|
line 273: participants who had "SOME" pre-existing |
Thanks for your suggestion. We put one or more pre-existing non-communicable chronic conditions. Please see page 10 line 295 |
|
line 275-278: this is in contrast with the next phrase line 278-281: this is in contrast with the next phrase |
Thanks very much for noting these discrepancies. We have revised these lines as follows: “Our study revealed that increase in tobacco use was significantly lower among the participants aged 70 years or less compared to their younger counterparts. We also found that the participants who had any pre-existing non-communicable and/or chronic conditions had less chance of increased tobacco use during the pandemic. This is possibly because relatively older participants and those who were suffering from any non-communicable and/or chronic conditions were more concerned of the deadliest effect of COVID-19, and thus had lower tobacco use”. Please see page 13 line 372-378. |
|
line 291-292: it could be explained by different income and level of literacy in rural and urban areas |
Thanks for your comment. We have added this in the revised manuscript. Please see page 12 line 347-351 |
|
line 296-298: why did you provide this information here? |
Thanks for noting this. We have removed this. Please see page 12 line 346-347 |
|
line 344: another strong limitation is that all information are self-reported so are based on subjective perceptions |
Thanks for noting this important limitation of the study. We have included this. Please see page 14 line 424-428 |
|
line 346-347: this conclusion is not well supported by results (you can only describe association not causality), I sugget to modify |
We have revised the Conclusion section. |
|
line 369: add that the consent was verbal |
Thank you, added. |
Reviewer 4 Report
The present manuscript titled "Changes in tobacco use pattern during COVID-19 and its correlates among older adults in Bangladesh" is an interesting article that deals with explored the changes in tobacco use pattern during COVID-19 pandemic and its correlates among the older adults in Bangladesh. It is a topical and interesting article. Present slight limitations that are detailed below:
- The use of uppercase n and lowercase n when dealing with the sample is not done in the correct way. You have to differentiate it.
- In the conclusions section, it would be convenient to add some more lines for the future. What are these results for?
- Some bibliographic references do not meet the journal's standards (eg reference 6, 22, 27 ...).
Author Response
|
Comment/Suggestion |
Authors’ Responses |
|
The present manuscript titled "Changes in tobacco use pattern during COVID-19 and its correlates among older adults in Bangladesh" is an interesting article that deals with explored the changes in tobacco use pattern during COVID-19 pandemic and its correlates among the older adults in Bangladesh. It is a topical and interesting article. Present slight limitations that are detailed below: |
|
|
1. The use of uppercase n and lowercase n when dealing with the sample is not done in the correct way. You have to differentiate it. |
Corrected in table 1. |
|
2. In the conclusions section, it would be convenient to add some more lines for the future. What are these results for? |
We have revised the Conclusion section. |
|
3. - Some bibliographic references do not meet the journal's standards (eg reference 6, 22, 27 ...). |
Thanks for noting this. We have checked every reference and corrected it where necessary in line with the journal guideline. |